# Space-time superoscillations

Yijie Shen ®[1,2] ✉, Nikitas Papasimakis ®[3] & Nikolay I. Zheludev ®[3,4]

Superoscillation (SO) refers to the phenomenon in which a wavefield locally oscillates at a rate exceeding its highest spatial or temporal Fourier component. SO has enabled light to be focused into arbitrarily small hotspots, forming the basis of superresolution imaging and metrology far beyond the Abbe-Rayleigh diffraction limit. Here we show that spatial and temporal superoscillations can occur simultaneously at the same point in space-time, a phenomenon we term space-time superoscillation (STSO). We demonstrate STSOs in a band-limited version of supertoroidal light pulses, a recently introduced family of space–time nonseparable finite-energy solutions of Maxwell's equations. Our results reveal a new regime of extreme spatio-temporal field structuring, with implications for ultrafast metrology, light–matter interactions, and deep-subwavelength control of electromagnetic waves.

Superoscillations (SOs) have emerged as a powerful concept in wave physics, enabling local oscillation rates that exceed the highest frequency present in the global spectrum[1,2]. This counterintuitive phenomenon—originally developed for waves in one, two, or three dimensions—has inspired applications in optical superresolution imaging, precision metrology, and structured light engineering[1–5]. The concept is also applicable to the temporal domain for ultrafast spectroscopy, and light–matter interactions[6,7].

In parallel, new phenomena have been observed in space-time nonseparable structured light fields[8–10]. Among them are supertoroidal pulses (STPs), a class of solutions to Maxwell's equations of rich topological structure[11]. A basic member of this family is the "Flying Doughnut" pulse—an experimentally demonstrated toroidal few-cycle waveform with tightly confined energy and nontrivial topology[12–15].

Here we show that the band-limited version of STPs include regions where both the local spatial wavevector and local frequency of oscillation in time exceed the respective global spectral bounds. We term this phenomenon space–time superoscillation (STSO).

## Results

### Band-limited supertoroidal pulses

An STP is characterized by two length parameters, $q_1$ and $q_2$, related to the central wavelength of the wave packet and the Rayleigh range, respectively, and a real dimensionless parameter $\alpha$, which controls the

pulse topology[11]. Finite-energy solutions require $\alpha \geq 1$; larger values of $\alpha$ yield increasingly rapid local field variations and stronger confinement[16]. STPs can be TE (transverse electric) or TM (transverse magnetic). Here we focus on the TE case without loss of generality. The full field expressions are provided as[11]:

$$E_\theta^{(\alpha)} = -2\alpha i f_0 \sqrt{\frac{\mu_0}{\varepsilon_0}} \left\{ \frac{(\alpha+1)r(q_1+i\tau)^{\alpha-1}(q_1+q_2-2ict)}{[r^2+(q_1+i\tau)(q_2-i\sigma)]^{\alpha+2}} - \frac{(\alpha-1)r(q_1+i\tau)^{\alpha-2}}{[r^2+(q_1+i\tau)(q_2-i\sigma)]^{\alpha+1}} \right\} \tag{1}$$

$$H_r^{(\alpha)} = 2\alpha i f_0 \left\{ \frac{(\alpha+1)r(q_1+i\tau)^{\alpha-1}(q_2-q_1-2iz)}{[r^2+(q_1+i\tau)(q_2-i\sigma)]^{\alpha+2}} - \frac{(\alpha-1)r(q_1+i\tau)^{\alpha-2}}{[r^2+(q_1+i\tau)(q_2-i\sigma)]^{\alpha+1}} \right\} \tag{2}$$

$$H_z^{(\alpha)} = -4\alpha f_0 \left\{ \frac{(q_1+i\tau)^{\alpha-1}[r^2-\alpha(q_1+i\tau)(q_2-i\sigma)]}{[r^2+(q_1+i\tau)(q_2-i\sigma)]^{\alpha+2}} + \frac{(\alpha-1)(q_1+i\tau)^{\alpha-2}(q_2-i\sigma)}{[r^2+(q_1+i\tau)(q_2-i\sigma)]^{\alpha+1}} \right\} \tag{3}$$

where $(r, \theta, z)$ are cylindrical coordinates, $t$ is time, $c = 1/\sqrt{\varepsilon_0\mu_0}$ is the speed of light, $\varepsilon_0$ and $\mu_0$ are the permittivity and permeability of vacuum, $\tau = z - ct$, $\sigma = z + ct$, and $f_0$ is a constant. In the cylindrical

[1]Centre for Disruptive Photonic Technologies, School of Physical and Mathematical Sciences, Nanyang Technological University, Singapore, Singapore. [2]School of Electrical and Electronic Engineering, Nanyang Technological University, Singapore, Singapore. [3]Optoelectronics Research Centre & Centre for Photonic Metamaterials, University of Southampton, Southampton, UK. [4]Institute for Advanced Study, Texas A&M University, College Station, TX, USA. ✉e-mail: yijie.shen@ntu.edu.sg

coordinate system, the instantaneous electric field can be written as $E_\theta(r,z,t) = A(r,z,t)e^{i\varphi(r,z,t)}$, where $A$ and $\varphi$ correspond to the amplitude and phase.

Although STPs are localized finite-energy solutions to Maxwell's equations, they are not band-limited. We therefore introduce a band-limited version of STPs, i.e. the STPs with their time domain and space domain spectra truncated at $f < f_m$ and $k < k_m$, respectively.

The entire spectrum of the STP is confined on the surface of light cone, i.e the conic surface with unit slope of its generatrix in the coordinate ($k_r$, $k_z$, $2\pi f/c$), where the spatial wavevectors are related to the spatial frequencies by $k_r = 2\pi f_r/c$ and $k_r = 2\pi f_z/c$. In the following, we will consider band-limited pulses:

$$\tilde{E}_c(r,z,t) = \int_{-\infty}^{\infty}\int_{-\infty}^{\infty}\int_{-\infty}^{\infty} \tilde{E}(f_r,f_z,f)B(f)\exp[i2\pi(f_r r + f_z z + f t)]J_0(2\pi r f_r)f_r df_r df_z df$$

(4)

where $\tilde{E}(f_r,f_z,f)$, the spectrum of a standard STP, is truncated by function $B(f)$ cut-off frequencies, $f_m$ and $k_m = 2\pi f_m/c$, and $J_0(2\pi rfr)$ is the 0-th order Bessel function of the first kind (see details in Supplementary Material S2). From now on, we will be analyzing the properties of this band-limited version of STP.

## Coexistence of spatial and temporal superoscillation

We illustrate the presence of STSO, i.e. a point in space-time where spatial and temporal SOs exist simultaneously, where both spatial (along the radial direction) and temporal phase gradients, $\partial\varphi/\partial r$ and $\partial\varphi/\partial t$, exceed $k_m$ and $f_m$ correspondingly, by considering a TE-mode band-limited version of an STP with $\alpha = 50$, $q_2 = 50q_1$, $q_1 = 1$ (see Fig. 1a for the electric field distribution $E_\theta(r,z,t)$ in space-time, and Fig. 1b for the field isosurface structure). The spatial (radial) variation of the electric field at ($z = 0$, $t = t_s$) is presented in Fig. 1c, where we observe that a segment in the off-axis region ($r = r_s$) oscillates considerably faster than the harmonic oscillation of the maximal radial frequency $k_m$ (dashed red line). Similarly, in Fig. 1d, we show the temporal variation of

the electric field at ($z = 0$, $r = r_s$), which around $t = t_s$ exhibits oscillations faster than frequency $f_m$ (dashed red line). Therefore, the field at the focal plane ($z = 0$) exhibits STSO.

We proove the existence of STSO of the band-limited STP in the ($r$, $t$) domain at focus ($z = 0$) for a fixed cut-off frequency $k_m = 2c/q_1$. In the SO region, the local temporal frequency or spatial wavevector should exceed the corresponding maximum value ($f_m$ or $k_m$)[1,17]. Figure 2a1–a4 shows $\log_{10}|\text{Re}[E_\theta(r,z=0,t)]|$, unwrapped phase of $\varphi(r,z=0,t)$, phase gradient, energy density and Poynting vector distributions, respectively, of the fundamental toroidal pulse ($\alpha = 1$). In Fig. 2a3, there is only a small region with temporal local wavevector $\partial\varphi/\partial t$ exceeding $f_m$, but no spatial SO and thus no STSO. In Fig. 2a4, we also highlight the region of energy backflow in accordance with prior works[11,18]. Higher values of $\alpha$ lead to increasingly complex pulses with multi-cycle structure and dramatic spatiotemporal evolution, which results into an extreme spatiotemporal focusing. Figure 2b1–b4 shows the corresponding characteristics of a band-limited STP at $\alpha = 50$. Figure 2b3 shows the presence of spatial SOs, where the local wavevector exceeds $k_m$, and temporal SOs, where the local temporal frequency exceeds $f_m$. Importantly, spatial and temporal SO regions overlap, resulting in STSOs. We can also observe that STSOs appear at low amplitude regions and are accompanied by areas of energy backflow, see inset to Fig. 2b4.

## Behaviors of supertoroidal pulses of different orders

Figure 3a, b shows temporal profiles of $\partial\varphi/\partial t$ at specific radius $r = 10q_1$ and spatial profiles of $\partial\varphi/\partial r$ at specific time $t = 2q_1/c$, respectively, for a series of band-limited STPs with different values of parameter $\alpha$. The spatial and temporal oscillations become stronger and faster with increasing $\alpha$. Figure 3c shows the ratio between the energy within the rapid oscillations region, $E_{STSO}$, and the total energy of the pulse, $E_T$, as a function of parameter $\alpha$. For $\alpha > 38$, the energy of STSOs monotonically increases with increasing $\alpha$, see Fig. 3c. In the SO region, the local temporal frequency or spatial wavevector should exceed the corresponding maximum value ($f_m$ or $k_m$). reaching values on the order of $10^{-3}$ of the total pulse energy— within reach of experimental detection[19–21].

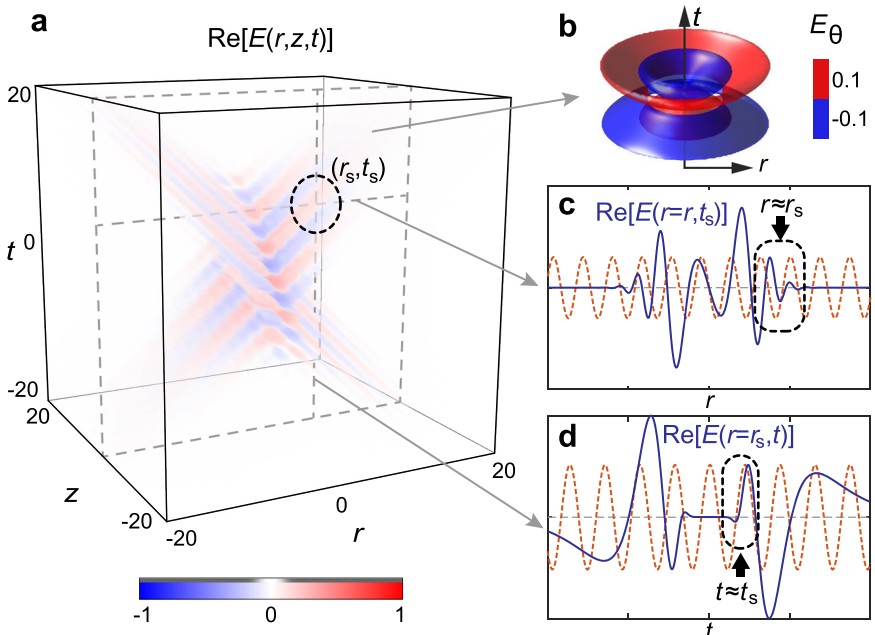

**Fig. 1 | Superoscillations in a band-limited supertoroidal light pulse ($\alpha = 50$, $q_2 = 50q_1$). a** Spatiotemporal evolution of the azimuthal electric field, $E_\theta(r,z,t)$. **b** Isosurfaces of the electric field, $E_\theta(r,z=0,t)$, in the focal plane. Radial (**c**) and temporal (**d**) field profiles at focus ($z = 0$) at a specific moment in time $t_s = 5$ (**c**, blue line), and at specific radial position $r_s = 10$ (**d**, blue line), respectively. The corresponding fastest spatial (**c**) and temporal (**d**) Fourier components are marked by red-dashed lines. The black dashed boxes highlight the SO region. Unit for length: $q_1$, Unit for time: $q_1/c$.

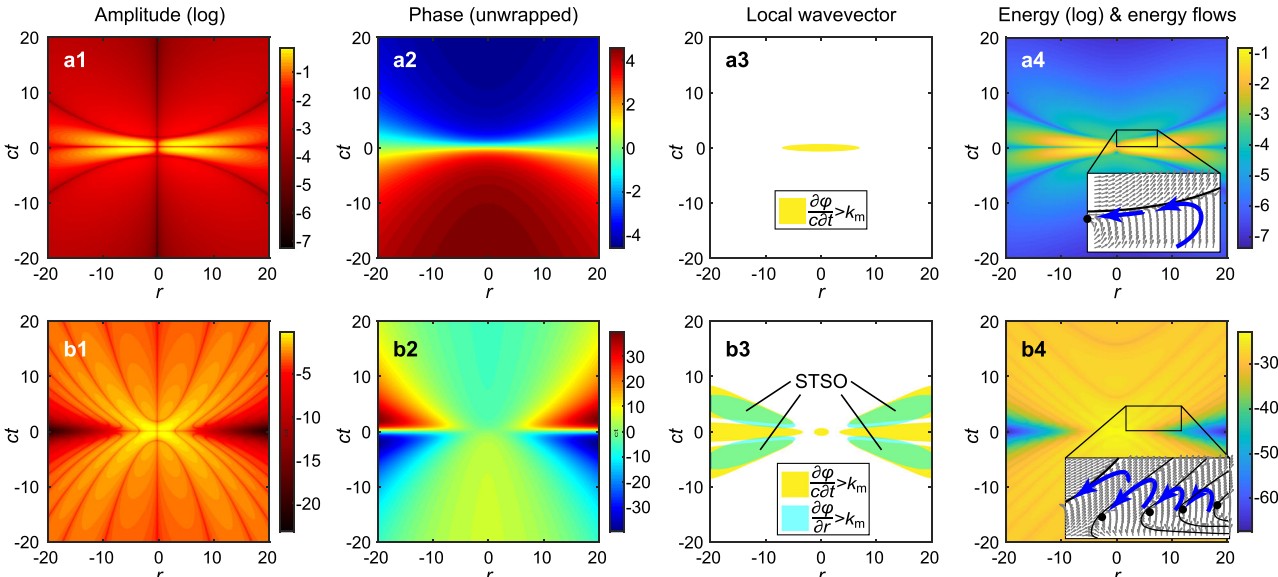

**Fig. 2 | Spatiotemporal structure of band-limited toroidal and supertoroidal pulses. a1, b1** Spatiotemporal field modulus ($|E_\theta(r,z=0,t)|$) distributions for a toroidal pulse ($\alpha=1$) (**a1**) and an STP ($\alpha=50$) (**b1**), at focus ($z=0$). The amplitude is presented in terms of the logarithm of its real part, $\log_{10}|\mathrm{Re}[E_\theta(r,z=0,t)]|$. **a2, b2** Unwrapped phase distributions $\varphi(r,t) = \mathrm{Arg}[E_\theta(r,z=0,t)]$ of the two pulses presented in (**a1, b1**). **a3, b3** Regions in which the radial local wavevector ($\partial\varphi/\partial r$) and local temporal frequency ($\partial\varphi/\partial t$) of the toroidal (**a3**) and supertoroidal (**b3**) pulse

exceed the threshold frequency, $f_m$, and wavevector, $k_m$, respectively ($k_m = 2/q_1$). **a4, b4** The spatial and temporal distribution of energy density, $w = (\epsilon_0 E^2 + \mu_0 H^2)/2$, for the two pulses. Insets show the local energy flow with the black solid lines and dots marking the zero lines and singularities, and blue thick arrows marking the areas of energy backflow. Unit for all axes, $r$ and $ct$, is $q_1$. Unit for phase is radians, while amplitude and energy density are in arbitrary units.

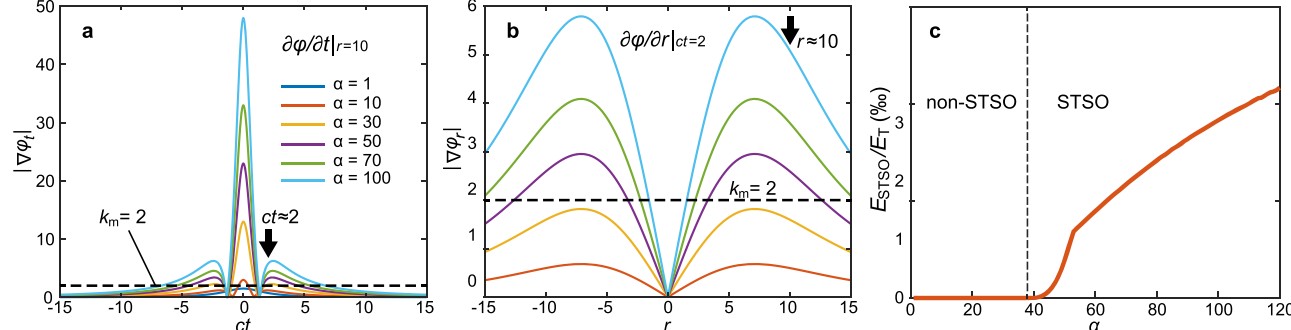

**Fig. 3 | Characterizing space–time superoscillations.** The temporal local frequency at $r=10$ (**a**) and the radial local wavevector at $t/c=2$ (**b**) of supertoroidal pulses for different values of $\alpha$. The black dashed lines mark the value of the fastest spatial and temporal frequency component, respectively. Units for $ct$ and $r$ are $q_1$.

**c** The ratio of energy in the STSO region ($E_{STSO}$) over the total energy ($E_T$) of the pulse as a function of supertoroidal order $\alpha$. The vertical dashed line at $\alpha = 38$ marks the onset of STSOs.

## Spectral signatures outside of the light cone

A hallmark of SO is that local spectra contain frequency components that exceed the global band limit of the entire field[1]. Similarly, we demonstrate that the spectra of STSO segments of band-limited STPs contain components out of the light cone, i.e. they oscillate faster than what would be expected based on the pulse bandwidth. Figure 4 shows the entire spectra (a1–a4) and local spectra (b1–b4) of band-limited STPs of different order, $\alpha = 1, 10, 50,$ and $100$. The local spectra are retrieved by the Fourier transform of the local space–time segment $E_\theta(r-r_s, z=0, t-t_s)$ into the $(k_r, f)$ domain. The spectra of full pulses are confined on the surface of the light cone, whereas local spectra present spectral components outside the light cone. The presence of the off-light-cone components directly reveals the presence of super-oscillations. Whereas for the fundamental toroidal pulse ($\alpha=1$), local spectra are fully contained within the light cone, in the case of the STPs ($\alpha>1$) the off-cone components become stronger with increasing value

of $\alpha$. Therefore, the field locally oscillates faster than permitted by their global spectra.

## Discussion

The STSOs in band-limited toroidal pulses shall be detectable across the electromagnetic spectrum. Optical, THz, and microwave platforms have already demonstrated generation of elementary toroidal pulses and complex hybrid toroidal pulses[13–15,21]. For an estimate in the optical domain, a Ti:Sa laser with a $\Delta\lambda = 200\,\mathrm{nm}$ bandwidth (FWHM) centered at $\lambda_0 = 800\,\mathrm{nm}$ yields spatial and temporal focusing limits of approximately $\lambda_0/2 = 400\,\mathrm{nm}$ and $\Delta\tau = \frac{K\lambda_0^2}{c\Delta\lambda} = 4.64\,\mathrm{fs}$ (constant $K = 0.441$ for Gaussian profile pulse), respectively. Based on the results in Fig. 3a, b, for an STP of $\alpha = 100$, at the STSO spot, the temporal and spatial local frequencies are about 7 and 5 times larger than the corresponding maximum values in the global spectra of the pulse, respectively, implying achievable hotspots temporal and

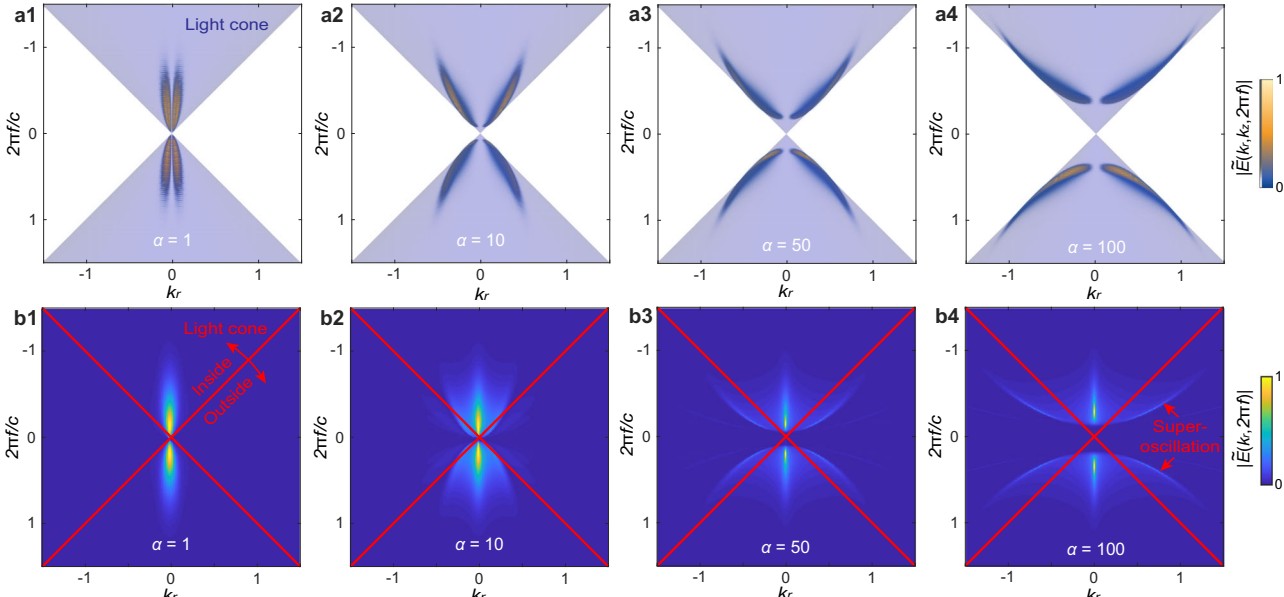

**Fig. 4 | Spectral representation of STSOs. a1–a4** Entire spectral power projected in the $(k_r, f)$ plane for toroidal pulses ($\alpha = 1$) and STPs ($\alpha = 10, 50, 100$). The light blue regions mark the light cone. **b1–b4** Local spectral power in the $(k_r, f)$ plane corresponding to radial-temporal STSO segments of pulses of different order ($\alpha = 1, 10,$ 50, 100). The red lines mark the boundary of the light cone. Note that, whereas the spectra of the full pulses (**a1–a4**) are confined on the surface of the light cone, local segments (**b1–b4**) exhibit out-of-cone components. Unit: $1/q_1$.

spatial confinements of about $\Delta\tau/7 = 0.66$ fs and $\lambda_0/2/5 = 80$ nm. Isolated temporal superoscillations can be even more extreme (see central peak of Fig. 3a), potentially enhancing confinement 50-fold.

The energy content of STSO regions (~0.1–1% of total energy, see Figs. 2b, 3c) is comparable to or higher than that successfully exploited in prior SO-enabled microscopy, metrology, and wavefront-shaping experiments[6,7,19,20]. For instance, in SO super-microscope experiments[19], it suffices that the SO hotspot contains only a small fraction (0.1%) of the total energy of the incident field, while high energy lobes are filtered out. Established techniques for detecting weak high-spatial-frequency features in the presence of strong sidelobes are therefore directly applicable.

Our findings demonstrate that simultaneous spatial and temporal superoscillations are physically realizable in finite-energy electromagnetic pulses. Moreover, one can superimpose a pulse chain of many STSOs to generate SO spatiotemporal arrays, analogous to previously demonstrated methods to produce multi-lobe SO arrays in the spatial domain[22,23].

We expect numerous applications of STSOs. In particular, spatial SOs have recently led to superresolution metrology even with picometric resolution[24,25], microscopy and imaging[1,19,20], while temporal SOs are finding applications in advanced spectroscopies[7]. Therefore, by combining these two forms of SOs into STSOs, we anticipate applications in metrology, imaging, sensing, and spectroscopy at ultrafine spatial and ultrafast temporal resolution[26]. Moreover, STSOs emerge in regions of nested skyrmionic topological structure[11], which can also be found in nondiffracting pulses[27]. Recently, optical skyrmions found applications in subwavelength sensing, metrology, and light–matter interaction[28–30], therefore, we anticipate that skyrmions with SO and STSO have potential to further enhance the deep-subwavelength resolution in these applications. Although demonstrated here for electromagnetic waves, the underlying mechanism of STSOs is universal and we anticipate that STSO behavior may be found in other spatiotemporally structured waves.

## Data availability
The data from this paper can be obtained from the University of Southampton ePrints research repository at https://doi.org/10.5258/

SOTON/D3797 and Nanyang Technological University data repository at https://doi.org/10.21979/N9/41ZJB1.

## Code availability
The code from this paper and details on the code used can be obtained from the University of Southampton ePrints research repository at https://doi.org/10.5258/SOTON/D3797 and Nanyang Technological University data repository at https://doi.org/10.21979/N9/41ZJB1.

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

## Acknowledgements

The UKs Engineering and Physical Sciences Research Council (grant EP/T02643X/1, Funder Id: 10.13039/501100000266), the European Research Council (Advanced grant FLEET-786851 & Proof of Concept Grant ASTRA-101248385, Funder Id: https://doi.org/10.13039/501100000781), the Defense Advanced Research Projects Agency (DARPA) under the Nascent Light Matter Interactions program, Singapore MOE AcRF Tier 1 grants (RG157/23 & RT11/23), Singapore Agency for Science, Technology and Research (A*STAR) MTC Individual Research Grants (M24N7c0080), and a Nanyang Assistant Professorship Start Up grant.

## Author contributions

Y.S, N.P. and N.I.Z. conceived the idea of the research. Y.S. performed the theoretical simulations and created graphical illustrations. All authors contributed to the manuscript preparation and discussion of results.

## Competing interests

The authors declare no competing interests.
