## [Transparent Peer Review file · Nature Communications]

Space-Time Superoscillation

Corresponding Author: Professor Yijie Shen

Version 0:

Reviewer comments:

Reviewer #1

(Remarks to the Author)

This work theoretically demonstrated a concept of space-time super-oscillation by introducing the super-toroidal pulse with non-separable nature in space and time domain. Through extends the concept of super-oscillation to the space-time domain, super-oscillation effect can be created at a special location in space and at a specific moment in time. This is a quite interesting work in science. However, I think it may not suitable for publication in this journal for the following reasons:

1. The content of the manuscript is heavily inclined towards physics, which do not align with the scope of this journal. While the theoretical insights into spatial-time super-oscillation are valuable, they are more suited to a specialized physics journal.

2. The manuscript presents a well-structured theoretical framework for spatial-time super-oscillation. However, it lacks experimental validation. Without experimental results, it is difficult to fully assess the validity and applicability of the proposed theories. I strongly suggest that the authors conduct relevant experiments to support their theoretical work. This would significantly enhance the credibility and value of their research.

3. Another major concern is the unclear application prospects of the research. We can see from the simulation results that the spectra of STSO segments of STPs contains components out of the light cone, and the spatial and temporal SOs overlap indicates the presence of STSOs. However, the STSO only exist at a specific space and time, and the extremely low amplitude also surrounded by a high amplitude field. How to use this physical effect in the practical scenarios. For a paper to be published in NC, it is essential that the potential applications of the research are clearly discussed, which could help readers understand the significance and relevance of the work.

In conclusion, I hope that the authors will consider these comments and revise their manuscript accordingly for future submission to a more suitable journal.

(Remarks on code availability)

Reviewer #2

(Remarks to the Author)

Comments for "Space-Time Superoscillations"

The authors presented the theoretical manner of constructing superoscillating field in space-time domain, by extending the model presented in ref. [11] in the main text, which, honestly, is an intriguing and promising subject, and probably is a key to precisely manipulate ultrafast pulse of light. This theory is introduced completely and lucidly, establishing the important criteria for identifying superoscillation, and also providing adequate details for who follows this work. However, the authors merely find the superoscillatory cases, rather than reveal the underlying primary phenomena which are different from the relevant object in other fields.

I strongly suggest to generalize this kind of spatiotemporal structures to have higher complexity, and more controllable degrees of freedom, e.g. the superoscillating spatiotemporal arrays [Opt. Lett. 45, 2538-2541 (2020); Nat. Commun. 14, 3237 (2023)].

The additional discussions uncovering the promising prospect on light-matter interaction is another way to increase the contents of this manuscript. I really look forward to the novel effects induced by the coupling between superoscillating pulse and new material, e.g. controlling the dynamics of phonons via the presented light field [Nat. Mater. 20, 607-611 (2021)].

Conventionally, superoscillating main lobes just occupy little energy, affected by the side-lobes. Here, the analysis of power occupation mentioned that the energy of superoscillating region obtains the higher ratio versus total energy, as index α increases, but more explanations were not given out.

According to my knowledge, the superoscillating signal can be designed by superimposing the signals with different parameters. Is it possible to construct the space-time superoscillation in this way (the units are replaced by the fields with different parameters α)? Can it improve the energy occupation of superoscillating lobes?

It is better to interpret the motivation of implementing the "Fourier transform" in Eq. (S21) of supplementary material, which is distinct to the conventional one, i.e. Hankel transform.

The important way to construct the bandlimited supertoroidal pulse should be introduced in the main text, rather than in supplementary material.

The group velocity, as an essential index, is not given out in the manuscript, which determines the propagation of supertoroidal pulse.

The shapes of superoscillating pulse ($\alpha=50$) is only exhibited in Fig. 1b, but the higher order cases were not given out in the appropriate figures.

Finally, this manuscript presents an attractive concept, but more primary effects induced by this supersocillation are not revealed here. It would be remarkable to add more contents to this manuscript.

(Remarks on code availability)

Version 1:

Reviewer comments:

Reviewer #1

(Remarks to the Author)

In the revised manuscript, the authors have added some new content to the Discussion section, as well as a section explaining the bandlimited spatiotemporal STP pulse. While certain reviewer questions have been addressed, the revisions do not introduce any essential improvements to the original version.

The authors claim that spatial and temporal superoscillations can be simultaneously observed at a specific point in space and time using spatiotemporally structured supertoroidal pulses, based on theoretical analysis. However, no experimental evidence is provided to support this claim. As shown in Figure 1, the spatiotemporal superoscillations (STSOs) occur only at a particular off-axis position and moment, and with very low amplitude. Under such conditions, how can this effect be experimentally validated?

Furthermore, the authors claimed that the bandwidth and pulse duration can be compressed by factors of 5 and 45, respectively, while applying STSO effect to a Ti:sapphire femtosecond laser. Yet, no detailed explanation is given regarding how this can be achieved, and no experimental data are presented. The feasibility and potential applications of this effect cannot be convincingly demonstrated based solely on calculated values. Similar concerns apply to the other proposed applications in metrology, imaging, and spectroscopy.

In summary, while the concept of spatiotemporal superoscillation is scientifically intriguing, a carefully designed experimental demonstration is essential to validate the proposed STSO effect. I strongly encourage the authors to devote effort on experimental verification. Without it, the manuscript may not meet the publication standards of a high-impact journal such as Nature Communication.

(Remarks on code availability)

Reviewer #2

(Remarks to the Author)

In the revised manuscript, the authors make more discussions on the potential applications (required by reviewer #1), and the more theoretical analysis, which has addressed my previous comments. I have to say again that this topic is interesting and promising. However, the revisions still cannot resolve some of my concerns. In order to demonstrate the importance of space-time superoscillation, I expect that the authors were able to unravel the potential phenomena with the use of presented theory, e.g. the intriguing dynamics considering both space and time domains rather than only introducing the mathematical framework. In my opinion, the new physical concept is closely associate with relevant essential effects, which, however, the authors did not particularly mention in the revision. Honestly, the used mathematical techniques are not surprising, which were mentioned in the previous papers. As a result, more contents of revealing particular physical phenomena are necessary. Otherwise, as suggested by the reviewer 1, experimental demonstrations on the space-time concept would be helpful. Without these, I agree with reviewer 1 that the manuscript may be suitable to a specialized journal,

rather than the Nat. Commun.

(Remarks on code availability)

None.

Reviewer #3

(Remarks to the Author)

The manuscript first proposed and systematically demonstrated the phenomenon of "space-time superoscillation", a groundbreaking concept in my opinion, which extends the concept of Michael Berry's superoscillation from the pure space or time domain to the spacetime joint domain, connecting to the recently emerging topic of spatiotemporal optics and physics. This work has significant theoretical innovation and potential application values. With very solid theories, the STSO phenomenon is clearly demonstrated and verified through multiple dimensions such as local wave vector/frequency, energy distribution, energy flow, and spectral analysis. The experimental plan and the numerical simulation part are reasonably designed and fully demonstrated. The article has a clear structure and smooth language, professionally written, undoubtedly meeting the high-level standards of Nature Communications. I recommend publication without delay. Some suggested minor revisions are as following:

1. The statement that "the group velocity of the STPs of different orders is always the constant of c " is a critical point. While the proof is correctly relegated to the supplement, its fundamental importance warrants a slightly more substantive note in the main text. Please consider adding a short justification, such as: "This is expected as STPs are vacuum solutions to Maxwell's equations and thus propagate without dispersion."
2. To enhance its depth, it would be valuable to briefly acknowledge the signal-to-noise challenge inherent in detecting STSOs, given their occurrence in low-field regions. A sentence could be added, for instance: "We note that the practical observation of STSOs, like other superoscillatory features, will require techniques sensitive to weak signals amidst higher-amplitude backgrounds, a challenge successfully overcome in prior spatial superoscillation measurements XXX."
3. In prior works, topological skyrmion textures were demonstrated in such STP pulses, I believe the authors may also need to discuss more on how the STSO associated with skyrmion textures can hatch more meaningful applications, for instance, deep-subwavelength sensing, metrology, and light-matter interaction.
4. "Abbey-Reileigh" should be the more common "Abbe-Rayleigh".
5. Unit writing: For example, "6 fs long pulses" is suggested to be changed to "6-fs-long pulses" or "pulses with a duration of 6 fs". It is suggested that " ~ 80 nm" be changed to " ≈ 80 nm" or "about 80 nm".
6. "In particular, the parameter α ..." The beginning of this sentence is a bit awkward. It is suggested to modify it to "The parameter α is a real dimensionless number that..." .
7. "Our work expands the concept of SO to include spatiotemporal effects, illustrated by the arbitrarily spatially and temporally confined focal spots in STPs. "this sentence is a bit long and can be split or reassembled for clarity.
8. The conclusion part uses two "Finally" in a row. It is suggested that the last "Finally" be changed to "Furthermore" or "Beyond electromagnetism".

(Remarks on code availability)

Comments from the reviewers:

Reviewer #1 (Comments to the Author):

This work theoretically demonstrated a concept of space-time super-oscillation by introducing the super-toroidal pulse with non-separable nature in space and time domain. Through extends the concept of super-oscillation to the space-time domain, super-oscillation effect can be created at a special location in space and at a specific moment in time. This is a quite interesting work in science. However, I think it may not suitable for publication in this journal for the following reasons:

Response: We thank the Reviewer for the positive comments. In the following we address their concerns.

1. The content of the manuscript is heavily inclined towards physics, which do not align with the scope of this journal. While the theoretical insights into spatial-time super-oscillation are valuable, they are more suited to a specialized physics journal.

Response: We respectfully disagree with the assertion that our work is not aligned with the scope of Nature Communications. As clearly stated on the journal's website: "Nature Communications is an open access, multidisciplinary journal dedicated to publishing high-quality research in all areas of the biological, health, **physical**, chemical, Earth, social, mathematical, applied, and engineering sciences. Papers published by the journal aim to represent **important advances of significance to specialists within each field.**"

Our work on space-time superoscillations represents a significant theoretical advancement in physics, with potential applications in various fields such as optics, quantum mechanics, and materials science. Our findings extend the concept of superoscillations to the space-time domain and will be of substantial interest to specialists within this research area. As such, they align well with the journal's commitment to publishing impactful research.

2. The manuscript presents a well-structured theoretical framework for spatial-time super-oscillation. However, it lacks experimental validation. Without experimental results, it is difficult to fully assess the validity and applicability of the proposed theories. I strongly suggest that the authors conduct relevant experiments to support their theoretical work. This would significantly enhance the credibility and value of their research.

Response: We agree that experimental validation of theoretical predictions is important. However, the purpose of the current manuscript is different: it seeks to draw attention to the unexpected phenomenon of space-time superoscillations in space-time nonseparable pulses. We argue that it is timely to present the theory of space-time superoscillations now, which can stimulate interest not only in the Optics community, but also by researchers working in acoustics, quantum mechanics, fluid dynamics.

We also note that space-time superoscillating pulses are exact solutions of Maxwell's equations, one of the most successful theories of classical physics. Thus, experimental work would target the practicality of generating space-time superoscillations, rather their validity.

3. Another major concern is the unclear application prospects of the research. We can see from the simulation results that the spectra of STSO segments of STPs contains components out of the light cone, and the spatial and temporal SOs overlap indicates the presence of STSOs. However, the STSO only exist at a specific space and time, and the extremely low amplitude also surrounded by a high amplitude field. How to use this physical effect in the practical scenarios. For a paper to be published in NC, it is essential that the

potential applications of the research are clearly discussed, which could help readers understand the significance and relevance of the work.

Response: The applications of STSOs should be clear and highly anticipated. The spatial superoscillation and temporal superoscillation have recently created advanced applications individually. For example, the spatial superoscillation has recently enabled ultra-small super-resolved nanometric metrology [Science364,771-775(2019)], microscopy and imaging [Nat. Rev. Phys. 4, 16–32 (2022), Nature Mater 11, 432–435 (2012)], even picometric metrology [Nat. Mater. 22, 844–847 (2023)]. In the case of temporal superoscillations, advanced spectroscopies have been proposed [Phys. Rev. Lett. 131, 153803 (2023)]. Therefore, by combining for the first time these two forms of superoscillations into space-time superoscillations, we anticipate advanced applications in metrology, imaging, sensing, and spectroscopy, at ultra-fine spatial and ultrafast temporal resolution.

We have added specific examples and references to our revised manuscript to illustrate these potential applications. Additionally, we have included a section on how space-time superoscillations can be used to enhance light-matter interactions, which could lead to novel applications in quantum technologies and material science. **See added content in the second-last paragraph in Discussion highlighted in the main text.**

In conclusion, I hope that the authors will consider these comments and revise their manuscript accordingly for future submission to a more suitable journal.

Response: We hope that we have convinced the reviewer that our work is suitable for publication in Nature Communications.

Reviewer #2 (Comments to the Author):

The authors presented the theoretical manner of constructing superoscillating field in space-time domain, by extending the model presented in ref. [11] in the main text, which, honestly, is an intriguing and promising subject, and probably is a key to precisely manipulate ultrafast pulse of light. This theory is introduced completely and lucidly, establishing the important criteria for identifying superoscillation, and also providing adequate details for who follows this work.

Response: We thank the referee for the very positive comments.

However, the authors merely find the superoscillatory cases, rather than reveal the underlying primary phenomena which are different from the relevant object in other fields.

Response: We thank the reviewer for raising this important issue. We would like to stress that the space-time superoscillatory pulses presented in our manuscript have no analog in other fields; this is the first illustration of space-time superoscillations (STSOs). Further, our work invites key questions about the underlying physics of electromagnetic STSOs. For instance, what is the role of space-time nonseparability in the existence of STSOs? Is there a canonical route to the construction of STSOs? We discuss these questions in the second paragraph in Discussion highlighted in the main text.

I strongly suggest to generalize this kind of spatiotemporal structures to have higher complexity, and more controllable degrees of freedom, e.g. the superoscillating spatiotemporal arrays [Opt. Lett. 45, 2538-2541 (2020); Nat. Commun. 14, 3237 (2023)].

Response: Thank you for your suggested references, they are indeed useful, we have cited them in our updated version. The STSO spots demonstrated in our manuscript are in isolated form, which are possible to be extended into more complex structured form, such as STSO spot array and multi-lobe forms. For instance, we can linearly superimpose a pulse chain of many STSOs to generate SO spatiotemporal arrays, analogous to the previous methods to produce multi-lobe SO arrays in the spatial domain [Opt. Lett. 45, 2538-2541 (2020); Nat. Commun. 14, 3237 (2023)]. See the end of the second paragraph in Discussion.

The additional discussions uncovering the promising prospect on light-matter interaction is another way to increase the contents of this manuscript. I really look forward to the novel effects induced by the coupling between superoscillating pulse and new material, e.g. controlling the dynamics of phonons via the presented light field [Nat. Mater. 20, 607-611 (2021)].

Response: Thank you for your valuable suggestion and suggested reference. Indeed, we believe the STSO in light pulses will unlock nontrivial light-matter interaction, we added the discussion and cited the suggested reference. See the end of the second-last paragraph in Discussion.

Conventionally, superoscillating main lobes just occupy little energy, affected by the side-lobes. Here, the analysis of power occupation mentioned that the energy of superoscillating region obtains the higher ratio versus total energy, as index α increases, but more explanations were not given out.

Response: We thank the reviewer for their valuable suggestion. The physical meaning of parameter α is to ensure finite energy localization of the pulse (see original work by

Ziolkowski: “Localized transmission of electromagnetic energy”, Phys. Rev. A 39, 2005 (1989)). In particular, the parameter α is a real dimensionless number that must satisfy $\alpha \geq 1$ to ensure finite energy solutions, while $\alpha < 1$ results in pulses of infinite energy, such as planar waves and cylindrical waves. With increasing α , the local oscillations become increasing faster leading to stronger confinement. However, the total energy contained in the superoscillatory region remains relatively small (~3% or lower) in accordance with conventional superoscillations. See the added explanation and reference in the first and next-to-last paragraphs of Results highlighted in the main text.

According to my knowledge, the superoscillating signal can be designed by superimposing the signals with different parameters. Is it possible to construct the space-time superoscillation in this way (the units are replaced by the fields with different parameters α)? Can it improve the energy occupation of superoscillating lobes?

Response: Thank you for your valuable insights. Indeed, the space-time superoscillatory pulses can be constructed by an appropriate superposition of plane waves as presented in Fig. 4a of the manuscript and Figure S2 of the Supplementary Materials. It is also likely that a different basis may be more convenient (especially a cylindrically symmetric one) allowing to maximize the energy of the superoscillatory lobes, a question that we reserve for future work. This is now discussed in paragraph 3 of the Discussion section.

It is better to interpret the motivation of implementing the “Fourier transform” in Eq. (S21) of supplementary material, which is distinct to the conventional one, i.e. Hankel transform.

Response: We thank the reviewer for this question. Since the monochromatic components of the pulse lie on the light cone, the temporal Fourier transform is sufficient.

The important way to construct the bandlimited superoscillating pulse should be introduced in the main text, rather than in supplementary material.

Response: Thanks for the suggestion. We have modified the main text accordingly. See the second paragraph of Results.

The group velocity, as an essential index, is not given out in the manuscript, which determines the propagation of superoscillating pulse.

Response: Theoretically, the group velocity of the STPs of different orders is always the constant of c , see detailed proof in Supplementary Materials S1 around Eq. (S20) We add explanation in the end of first paragraph of Results.

The shapes of superoscillating pulse ($\alpha=50$) is only exhibited in Fig. 1b, but the higher order cases were not given out in the appropriate figures.

Response: We now include higher order cases in Fig. S1 of the supplementary material.

Finally, this manuscript presents an attractive concept, but more primary effects induced by this superoscillation are not revealed here. It would be remarkable to add more contents to this manuscript.

Response: We thank the referee for the very positive comments.

Response to reviewers' comments

Reviewer #1 (Remarks to the Author):

In the revised manuscript, the authors have added some new content to the Discussion section, as well as a section explaining the bandlimited spatiotemporal STP pulse. While certain reviewer questions have been addressed, the revisions do not introduce any essential improvements to the original version.

Response: We thank the reviewer for acknowledging our revision. We believe we have addressed all comments raised in the previous round of review. We also address below all comments raised in the current round.

1. The authors claim that spatial and temporal superoscillations can be simultaneously observed at a specific point in space and time using spatiotemporally structured superoscillatory pulses, based on theoretical analysis. However, no experimental evidence is provided to support this claim. As shown in Figure 1, the spatiotemporal superoscillations (STSOs) occur only at a particular off-axis position and moment, and with very low amplitude. Under such conditions, how can this effect be experimentally validated?

Response: We argue that the novelty of the work is sufficient to warrant publication in Nature Communications and expect that it will stimulate experimental investigations of STSOs. We have extended our discussion on the practical considerations of generating STSOs in the last paragraph of first subsection in Discussion.

Regarding “very low amplitude” issue, we would like to note that the amplitude of the STSOs is of the order 10^{-3} of the peak amplitude in a non-optimized case (see Fig. 2b), while the total energy contained in the STSO regions can exceed 0.3% (see Fig. 3c). This is significantly higher than what is typically encountered in conventional superoscillations and lies within current experimental capabilities. For instance, in superoscillatory microscopy [Sci. Rep. 3, 1715 (2013)], the SO hotspot is about 0.1% energy of the total energy. We include this discussion in the first section of Discussion.

2. Furthermore, the authors claimed that the bandwidth and pulse duration can be compressed by factors of 5 and 45, respectively, while applying STSO effect to a Ti:sapphire femtosecond laser. Yet, no detailed explanation is given regarding how this can be achieved, and no experimental data are presented. The feasibility and potential applications of this effect cannot be convincingly demonstrated based solely on calculated values. Similar concerns apply to the other proposed applications in metrology, imaging, and spectroscopy.

Response: We thank the reviewer for raising this point. We now describe in detail the calculation of spatial and temporal compression — “For instance, in the optical domain, commercially available Ti:Sa lasers can provide up to $\Delta\lambda = 200$ nm bandwidth (FWHM) central at $\lambda_0 = 800$ nm. The corresponding limit to spatial and temporal focusing are $\lambda_0/2 = 400$ nm and $\Delta\tau = \frac{K\lambda_0^2}{c\Delta\lambda} = 4.64$ fs (where $K=0.441$ for Gaussian profile pulse), respectively. Based on the results shown in Figs. 3(a) and 3(b), for STP of $\alpha=100$, at STSO spot, the temporal and spatial local

wavevectors are 7 times and 5 times larger than the central wavevector, respectively. Here, we evaluate central wavevector as $k_c=1$ (unit: $1/q_1$) as the central wavevector, it actually should be less than 1, see Fig. 4a. Therefore, we evaluate such STSO in STP allow temporal and spatial confinements of $\sim\Delta\tau/7=0.66$ fs and $\sim\lambda_0/2/5 = 80$ nm, respectively. If we apply the temporal SO separately (the central peak of Figs. 3(a) over 45 times than $k_c=1$), the temporal confinements can also be ~ 45 times shorter than $\Delta\tau$ (about 0.1 fs).” We replaced the corresponding statement into the more accurate in Discussion.

Our results indicate that very rapid local spatial and temporal field fluctuations are possible, which could find applications in metrology, imaging, and spectroscopy. We agree with the reviewer that the experimental verification is important and constitutes the next step forward.

In summary, while the concept of spatiotemporal superoscillation is scientifically intriguing, a carefully designed experimental demonstration is essential to validate the proposed STSO effect. I strongly encourage the authors to devote effort on experimental verification. Without it, the manuscript may not meet the publication standards of a high-impact journal such as Nature Communication.

Response: We refer the reviewer to our response to his previous point 1.

Reviewer #2 (Remarks to the Author):

In the revised manuscript, the authors make more discussions on the potential applications (required by reviewer #1), and the more theoretical analysis, which has addressed my previous comments. I have to say again that this topic is interesting and promising.

Response: We thank the reviewer for their supportive comments.

However, the revisions still cannot resolve some of my concerns. In order to demonstrate the importance of space-time superoscillation, I expect that the authors were able to unravel the potential phenomena with the use of presented theory, e.g. the intriguing dynamics considering both space and time domains rather than only introducing the mathematical framework. In my opinion, the new physical concept is closely associate with relevant essential effects, which, however, the authors did not particularly mention in the revision. Honestly, the used mathematical techniques are not surprising, which were mentioned in the previous papers. As a result, more contents of revealing particular physical phenomena are necessary.

Response: The reviewer raises the question of the necessary and sufficient physical conditions for STSOs to emerge. While we recognize the importance of this question, it lies beyond the scope of the current work that aims to highlight the existence of STSOs in space-time nonseparable light fields. A brief discussion is included in the last paragraph of Discussion, first paragraph of *Future perspectives*, in the main text.

Otherwise, as suggested by the reviewer 1, experimental demonstrations on the space-time concept would be helpful. Without these, I agree with reviewer 1 that the manuscript may be suitable to a specialized journal, rather than the Nat. Commun.

Response: We argue that the novelty of the work is sufficient to warrant publication in Nature Communications and expect that it will stimulate experimental investigations of STSOs. We also note here the scope of Nature Communications: "Papers published by the journal aim to represent important advances of significance to specialists within each field".

Reviewer #2 (Remarks on code availability):

None.

Reviewer #3 (Remarks to the Author):

The manuscript first proposed and systematically demonstrated the phenomenon of "space-time superoscillation", a groundbreaking concept in my opinion, which extends the concept of Michael Berry's superoscillation from the pure space or time domain to the spacetime joint domain, connecting to the recently emerging topic of spatiotemporal optics and physics. This work has significant theoretical innovation and potential application values. With very solid theories, the STSO phenomenon is clearly demonstrated and verified through multiple dimensions such as local wave vector/frequency, energy distribution, energy flow, and spectral analysis. The experimental plan and the numerical simulation part are reasonably designed and fully demonstrated. The article has a clear structure and smooth language, professionally written, undoubtedly meeting the high-level standards of Nature Communications. I recommend publication without delay.

Response: We thank the reviewer for their supportive comments.

Some suggested minor revisions are as following:

1. The statement that "the group velocity of the STPs of different orders is always the constant of c " is a critical point. While the proof is correctly relegated to the supplement, its fundamental importance warrants a slightly more substantive note in the main text. Please consider adding a short justification, such as: "This is expected as STPs are vacuum solutions to Maxwell's equations and thus propagate without dispersion."

Response: We thank the reviewer for the insightful comment. We revised the sentence into "In the condition of propagation in vacuum without dispersion, the group velocity on axis ($r=0$) of the STPs of different orders is always the constant of c , see Supplementary Materials S1, while the velocity of the centroid of the whole structured pulse is less than c ." See revised statement in the paragraph after Eq.(3).

2. To enhance its depth, it would be valuable to briefly acknowledge the signal-to-noise challenge inherent in detecting STSOs, given their occurrence in low-field regions. A sentence could be added, for instance: "We note that the practical observation of STSOs, like other superoscillatory features, will require techniques sensitive to weak signals amidst higher-amplitude backgrounds, a challenge successfully overcome in prior spatial superoscillation measurements XXX."

Response: In response to the reviewer's comment, we added the following statement in page XX, paragraph XX: "Moreover, the practical observation of STSOs, like other superoscillatory features, will require techniques sensitive to weak signals amidst higher-amplitude backgrounds, a challenge successfully overcome in prior spatial superoscillation measurements [6,7,Sci. Rep. 3, 1715 (2013)]." See the end of third paragraph of first subsection of Discussion.

3. In prior works, topological skyrmion textures were demonstrated in such STP pulses, I believe the authors may also need to discuss more on how the STSO associated with skyrmion textures can hatch more meaningful applications, for instance, deep-subwavelength sensing, metrology, and light-matter interaction.

Response: We thank the reviewer for their valuable comments. Indeed, the STSO phenomena are linked to the skyrmionic field structure. For instance, STSOs emerge in regions of rapid field reversal resulting from a nested skyrmionic topological structure [11], which can also be extended with nondiffracting robust propagation [26]. Recently, optical skyrmions found their applications in subwavelength sensing, metrology, and light-matter interaction [27-29], therefore, we perspective the skyrmions with SO and STSO has potential to further enhance the deep-subwavelength resolution in these applications. We have added the discussion in the second last paragraph of the Discussion.

4. "Abbey-Reileigh" should be the more common "Abbe-Rayleigh".

5. Unit writing: For example, "6 fs long pulses" is suggested to be changed to "6-fs-long pulses" or "pulses with a duration of 6 fs". It is suggested that "~80 nm" be changed to "≈80 nm" or "about 80 nm".

6. "In particular, the parameter a a..." The beginning of this sentence is a bit awkward. It is suggested to modify it to "The parameter a is a real dimensionless number that..." .

7. "Our work expands the concept of SO to include spatiotemporal effects, illustrated by the arbitrarily spatially and temporally confined focal spots in STPs. "this sentence is a bit long and can be split or reassembled for clarity.

8. The conclusion part uses two "Finally" in a row. It is suggested that the last "Finally" be changed to "Furthermore" or "Beyond electromagnetism".

Response: We have cleared up these minor issues.